# Culturomics Approach to Identify Diabetic Foot Infection Bacteria

**DOI:** 10.3390/ijms22179574

**Published:** 2021-09-03

**Authors:** Michał Złoch, Ewelina Maślak, Wojciech Kupczyk, Marek Jackowski, Paweł Pomastowski, Bogusław Buszewski

**Affiliations:** 1Centre for Modern Interdisciplinary Technologies, Nicolaus Copernicus University in Toruń, Wileńska 4 Str., 87-100 Toruń, Poland; ewelina.maslak@poczta.onet.pl (E.M.); p.pomastowski@umk.pl (P.P.); bbusz@umk.pl (B.B.); 2Department of General, Gastroenterological and Oncological Surgery, Faculty of Medicine, Collegium Medicum, Nicolaus Copernicus University in Toruń, Gagarina 7, 87-100 Torun, Poland; kupczykwojciech@cm.umk.pl (W.K.); marek.jackowski@cm.umk.pl (M.J.); 3Chair of Environmental Chemistry and Bioanalytics, Faculty of Chemistry, Nicolaus Copernicus University in Toruń, Gagarina 7 Str., 87-100 Toruń, Poland

**Keywords:** diabetic foot infection, culturomics, MALDI-TOF MS, bacteria

## Abstract

The main goal of the study was to evaluate the usefulness of the culturomics approach in the reflection of diabetic foot infections (DFIs) microbial compositions in Poland. Superficial swab samples of 16 diabetic foot infection patients (Provincial Polyclinical Hospital in Toruń, Poland) were subjected to culturing using 10 different types of media followed by the identification via the matrix-assisted laser desorption ionization-time of flight mass spectrometry (MALDI-TOF MS) and Biotyper platform. Identified 204 bacterial isolates representing 18 different species—mostly *Enterococcus faecalis* (63%) and *Staphylococcus aureus* (44%). Most of the infections (81%) demonstrated a polymicrobial character. Great differences in the species coverage, the number of isolated Gram-positive and Gram-negative bacteria, and the efficiency of the microbial composition reflection between the investigated media were revealed. The use of commonly recommended blood agar allowed to reveal only 53% of the entire microbial composition of the diabetic foot infection samples, which considerably improved when the chromagar orientation and vancomycin-resistant enterococi agar were applied. In general, efficiency increased in the following order: selective < universal < enriched < differential media. Performed analysis also revealed the impact of the culture media composition on the molecular profiles of some bacterial species, such as *Corynebacterium striatum*, *Proteus mirabilis* or *Morganella morganii* that contributed to the differences in the identification quality. Our results indicated that the culturomics approach can significantly improve the accuracy of the reflection of the diabetic foot infections microbial compositions as long as an appropriate media set is selected. The chromagar orientation and vancomycin-resistant enterococi agar media which were used for the first time to study diabetic foot infection microbial profiles demonstrate the highest utility in the culturomics approach and should be included in further studies directed to find a faster and more reliable diabetic foot infection diagnostic tool.

## 1. Introduction

The DFI is becoming a serious global life threat due to the increasing population of diabetes among which nearly 15–20% will suffer from a diabetic foot ulcer (DFU) during their lifetime [1,2]. A key factor in the effective management of the DFIs is a comprehensive empirical antimicrobial therapy that covers the most probable causative agents. Although international guidelines developed by e.g., the IDSA (Infectious Diseases Society of America, Arlington, VA, USA) and the IWDGF (the International Working Group on the Diabetic Foot) have become a vital instrument in choosing such treatments, there is a growing need for reports on the DFI microbial compositions in specific geographical regions to provide local treatment guidelines since microbiological studies of the DFIs conducted so far have yielded inconsistent results [3,4].

The accurate bacteria identification and diagnosing DFI in diabetic patients is still challenging due to usually a polymicrobial nature of the infection and the confounding effect of neuropathy and ischemia on the local and systemic inflammatory response [5]. Moreover, as with other skin wound infections, there are difficulties to define the pathogenic character of the microbial species involved due to the presence of a great number of commensal ones which, however, might become pathogenic when the opportunity arises [6].

Currently, several different approaches are used to identify and define the human microbiota. Mostly, the diagnosis in clinical microbiology relies on the phenotypic identification that consists in culturing microorganisms to grow and isolate their colonies [7]. However, the traditional culture method does not allow for the identification of new bacteria and those presented atypical phenotypical profiles. Such limitations can be overcome using the 16S rDNA gene sequencing which paved the way to identify rare, fastidious, and new microorganisms [8,9]. Recent studies indicated that the application of molecular techniques, such as the 16S rDNA PCR amplification, enabled to identify a greater diversity of the DFI microbiota including fastidious anaerobes and Gram-negative species than the standard culture methods [5]. Due to significant advances in the DNA sequencing techniques such as metagenomics which enable to investigate the microbial composition based on the genetic material directly recovered from the sample, many scientists started to believe that culture would no longer be needed [7,10]. Nevertheless, the microbial identification via molecular techniques faces several obstacles such as difficulties to define specific phenotypic characteristics for new species due to the limited number of available biochemical tests, overlooking minority species, the inability to distinguish between live/dead cells or the inability to assess antibiotic susceptibility [7,11]. Moreover, in the case of metagenomics, pure cultures of microorganisms are not provided and further strain characterization including host-interactions is not achievable [12].

As many studies have proven, culturing is still essential to describe new prokaryotic species and filling metagenomics gaps [13]. It is possible due to the use of fast and cost-effective bacterial identification by MALDI–TOF MS along with the multiplication of culture conditions called the culturomics approach [14]. Such a complementary strategy to metagenomics and the 16S rDNA analysis allows for a very rapid bacterial identification and description of the microbiological background of the disease; it also led to the discovery of hundreds of new human-associated bacterial species [15,16]. Culturomics was introduced in order to optimize culture conditions and revealed the potential to dramatically increase the number of bacteria identified by culture methods causing scientists to revise the term “unculturable organisms” since all microorganisms, following the statements of Bilen and his colleagues, are cultivable providing that the right conditions and tools are supplied [6,13]. The establishment of different culture conditions dedicated to a specific clinical specimen, including those that suppress the culture of majority populations and improve the growth of fastidious microorganisms, which could be further subjected to the rapid identification via the MALDI analysis (in less than one hour) is the crucial step of this approach [17,18]. The ability to isolate rather than simply obtain sequences of bacterial species is a big advantage of culturomics since it allows the further investigation of biological significance, features, and therapeutic potentials including antimicrobial susceptibility [13,18].

The growth conditions applied in the laboratory which lie at the heart of the culturomics approach, are the most important aspect of isolating and cultivating microorganisms, therefore, it is believed that the transition from the undefined to the defined set of media is crucial to enable interlaboratory comparisons and obtain reproducible results [19]. In most of the studies hitherto conducted on the DFI microbiota, culture conditions are not under debate and their selection is limited to the commonly accepted recommendations. There is a large variation in the type of the culture media used by researchers—from a single medium (e.g., Luria Bertani [20]) to several ones (e.g., Columbia blood, Trypticase soya, Chapman, M17, Eosin methylene blue, Hektoen and Cetrimide agars [21]) which may contribute to significant discrepancies in the identification results noted. Therefore, the main goal of the study was to evaluate the relative abilities of different media to recover bacterial isolates from DFI patients in Poland using the culturomics approach. Moreover, the influence of the type of medium on the composition of bacterial molecular profiles, and thus on the reliability of identification results, was discussed. In the research, commercially available culture media were used, which may facilitate the implementation of the established method in routine clinical laboratories in the future.

## 2. Results

### 2.1. Microbiological Background of Diabetic Foot Infection Samples

The applied MALDI analysis parameters allowed to obtain MS spectra of protein extracts of all tested bacterial isolates (exemplary spectra are presented in Figure 1), which were then used for identification via the MALDI Biotyper Compass Explorer 4.1 platform and BDAL database—updated version 1.0.16.0 with 6,903 Main Spectra (MSP) entries (27 June 2017). Detailed information about obtained score values depending on the culture media used are presented in Appendix A.

As a result of the microbiological composition analysis of the superficial swab samples obtained from the 16 DFI patients received 204 bacterial isolates representing 18 different species—8 Gram-negative and 10 Gram-positive—belonging to 3 types of bacteria: Firmicutes (50.0%), Proteobacteria (44.4%), and Actinobacteria (5.6%) (Figure 2).

Streptococci and staphylococci were the highly dominant group among all isolated bacterial strains. Most of the identified bacterial species represented facultative anaerobes, of which more than 80% are part of the microbiome of the gastrointestinal tract, skin, or respiratory tract. Nine of all identified bacterial species (50%) occurred in at least two patients, while another 9 were characteristic for individual patients. Most patients were affected by polymicrobial infections (81%)—from 2 to 5 different species at the same time (Figure 2). Most often, 4 different bacterial species were present in diabetic foot wounds (in the case of ca. 37% patients). The highest number of species was observed in patient P4, while samples derived from three patients were classified as monobacterial—only *Staphylococcus aureus* presence in each case.

Gram-positive bacteria were the most commonly isolated type of bacteria (15 out of 16 patients) and in the case of 50% samples were the only type of bacteria presented in the samples. Gram-negative bacteria were much less common—among 8 examined patients. The most common Gram-positive species were *Enterococcus faecalis* (10 cases, 63%), *Staphylococcus aureus* (7 cases, 44%) and *Corynebacterium striatum* (3 cases, 19%). Among the Gram-negative bacteria, *Pseudomonas aeruginosa* (4 cases, 25%) along with *Escherichia coli*, *Morganella morganii*, and *Proteus mirabilis* were the most frequently isolated—each strain was presented in the samples of 3 patients (19%).

### 2.2. DFI Species Coverage by Culture Media

All Gram-positive and Gram-negative species were isolated on all media except AZI. In 9 out of 10 media Gram-positive isolates constituted the majority—from 58% (COL) to 100% (AZI) (Table 1).

On one medium—CHRA—the percentage of Gram-negative bacteria was slightly higher than Gram-positive—52%. Considering the most dominant Gram-positive species, *S. aureus* demonstrated the highest percentage on MHA, BHI, and MAN medium, *E. faecalis* on CHRA, AZI, and VRE, while *C. striatum* on BLA medium. Regarding dominant Gram-negative species, *P. aeruginosa* were mostly isolated on TSA, *E. coli* and *M. morganii* on CHRA, while *P. mirabilis* on MAN, BLA, and BHI.

### 2.3. Evaluation of the Culture Media Usefulness for the Investigation of DFI Microbial Compositions

The analysis revealed great differences between tested culture media in terms of the number of isolated Gram-positive and -negative bacterial species (Table 2).

The largest number of different Gram-positive species was isolated on BLA medium (9 out of 10 species), universal COL medium, and selective VRE medium- 8 species for both. The lowest number of Gram-positive species were noted for AZI (5) and MAN (4) media. In the case of Gram-negative, the use of the CHRA medium enabled to detect all identified Gram-negative species (8). A slightly lower number was observed for BCP (7) and BLA medium (6), while the use of AZI medium completely inhibited the growth of this bacteria type according to its specification. In total, the most species were isolated on BLA and CHRA media (15 out of all 18 species), and the least on selective media: MAN and AZI (5 species each).

Considering the total number of different isolates derived from all samples tested, the lowest number of Gram-positive isolates was noted for MAN (8), while the highest for VRE (19). In the case of Gram-negative bacteria, the most isolates were obtained using CHRA medium (14), and the least on MAN (2) and AZI (0). In total, the largest number of bacterial isolates were obtained on the medium CHRA (28), VRE (25), and BLA (24), and the least on AZI (16) and MAN (10).

Taking into account the percentage of the reflected microbial profiles of the all tested DFI samples, the use of a single medium allowed to reveal up to 59% of the whole microbial compositions; however, in the case of half of the media, it did not exceed the value of 50% (Table 3).

The highest percentage was revealed for CHRA (55%), BCP (56%), and VRE (59%) medium. The use of a set of two media improved the reconstruction of the infection species composition by 17% on average and ranged from 60% to 73%. The highest values were reached for the media set containing CHRA or VRE—73% and 71% on average, respectively. When 3 media were simultaneously applied, the reliability of the reflection of the infections compositions improved by another 9% and ranged 72–81%. Similarly, CHRA and VRE media demonstrated the highest usefulness—81% and 78%, respectively.

### 2.4. Selection of the Most Effective Sets of Culture Media

The use of 10 different culture media resulted in 45 various two-component sets of culture media (Figure 3).

Considering individual media types, the average efficiency of the infections background reflection increased in the following order: selective (59%) < universal (61%) < enriched (68%) < differential (74%). The worst results were obtained for the set composed of two highly selective media—MAN and AZI—where the percentage of the reflected microbial composition of the investigated DFI samples did not even reach 50%. In contrast, the most useful media appeared to be CHRA and VRE in case of which their simultaneous use gave 82% coverage, which increased up to 88% when universal tryptic soy agar was additionally included.

## 3. Discussion

Applied culture conditions allowed to reveal the predominance of the Gram-positive species over the Gram-negative ones (59.6% to 40.4%) in the investigated samples. It confirms the common statement that Gram-positive species are most often isolated from DFI wounds in the developed European countries [22]. Similar observations were made for example in the work Jneid et al. [11] (France, 54.7%) or Dang et al. [23] (United Kingdom, 56.7% and 63.4% in 1998 and 2001, respectively). Besides the geographical location, the stage of infection has also an important impact on the microbial composition of the DFI wounds, since for patients with mild or early-stage infections Gram-positive bacteria were characterized as the predominate type [22]. Most of the investigated infections (81%) demonstrated the polymicrobial character (from 2 to 5 species at the same time). The nature of the DFI could be either mono- or polymicrobial, however, earlier studies have reported a higher frequency of polymicrobial ones [24,25] which may arise from the severity of the infections since severe ones exhibit a mostly higher percentage of polymicrobial patterns [26]. Nevertheless, the DFI samples investigated in our studies derived from patients affected by early stage or mild infections, therefore, the dominance of the polymicrobial infections could be explained by the notable participation of commensal skin microbiome rather than the severity of infections. This issue has been repeatedly shown for the DFI or other skin wound infections and still constitutes a key challenge in making a reliable diagnosis to apply an effective treatment [27].

Regarding identified bacterial species, all were previously reported in the literature as present during DFI development. Two species—*E. faecalis* and *S. aureus*—constituted the vast majority of all the identified isolates. *S. aureus* is considered to be the most common causative agent of the DFI among patients from Europe and North America, especially in the initial stages of the infection [28,29]. Moreover, *S. aureus* and other aerobic Gram-positive cocci are often reported as predominant pathogens in the monomicrobial DFIs [30], which was indicated in our study where among all monobacterial infections only *S. aureus* was detected. Such a phenomenon emerged as a substantial problem in view of the increasing appearance of methicillin-resistant *S. aureus* (MRSA) in DFIs observed in recent years [30]. Surprisingly, *E. faecalis* turned out to be the most dominant species. *E. faecalis* is found to be one of the most frequently isolated bacterial species in diabetic foot ulcer [31], however, the available literature fails to present studies proving this strain to be dominant except the work of Shettigar et al. [32] which reported similar frequency of the *E. faecalis* occurrence within the DFI samples—65%. It is believed that the presence of this bacteria in the wound contributes to the worsening of diabetic foot ulcers, especially during the polymicrobial infection, due to its capability to produce prolific amounts of biofilm and a high frequency of carrying drug-resistance genes which in the case of *E. faecalis* can be easily transferred to other bacterial species such as *S. aureus* [33,34].

Although the obtained results are in line with the previous European studies. One of the recent works by Michalek et al. [35]—also carried out in Poland—presented completely different observations. Among the 81 positive samples derived from 54 patients, the authors noted that the majority of the samples were monomicrobial (58.0%). Moreover, the share of Gram-positive species was only 22.9% while Gram-negative ones constituted a highly dominant group—77.1%. Additionally, significant differences in the species composition between our studies and the above-mentioned studies were observed. The most dominant species in our studies—*E. faecalis* (63%), accounted for only 2.5% identified isolates in the work by Michalek et colleagues [35]. Similarly, *P. aeruginosa* presented in a quarter of our samples constituted only 8.5% isolates in the above-mentioned work. Interestingly, despite considerable variations between the results of both studies, the percentage of *P. mirabilis* was very close—19% and 19.5% in our work and the work by Michalek et al., respectively, which has not yet been reported at such a level. This phenomenon may indicate that the results of the DFI microbial composition testing can vary significantly depending on the culture conditions used, even when hospitals are separated by a short geographic distance (~270 km in a straight line).

Our studies revealed big differences between the efficiency of culture media to reflect the DFI microbial composition resulted from their various species range coverage (see Table 1 and Table 3). Interestingly, the use of the single Columbia blood agar (BLA), which along with other blood enriched agars such as chocolate agar belongs to the most often used media for isolation of DFI microbiota [36,37,38,39], was sufficient to reflect only 53% of the entire microbiological profiles of the studied clinical cases and was lower compared to either differential media (CHRA and BCP) and selective VRE medium. However, the reflection of the DFI microbial compositions using BLA medium significantly improved when additionally CHRA and VRE media were applied—by 32%. Nowadays differential chromogenic agars (e.g., CHROMagar *Staph aureus*–CASA, MRSA ID) are becoming more popular than historical routine agars such as blood agar or mannitol salt agar in terms of *S. aureus* and MRSA detection within diabetic foot wounds [40]. One of the last studies on the Chromagar orientation was used to boost the discriminatory power of routine phenotypic identification protocols for UTI (Urinary tract infections) bacteria among diabetic patients attending clinics in Bushenyi district of Uganda [41], however, best of our knowledge there is a lack of studies about their application in the investigation of the entire DFI microbiota. The results obtained by us indicate the high utility of the Chromagar orientation in the investigation of DFI microbiota via the culturomics approach and fulfill a recent conclusion of Perry [42] that the combination of differential chromogenic media and MALDI-TOF MS analysis may compensate for any lack of specificity and may contribute to obtain results in an even shorter time. The same applied to selective VRE medium, which in combination with the CHRA gave almost the same efficiency in the reflection of the DFI microbial compositions as in the three-component media set including also the commonly used Columbia blood agar.

Microbial identification via the MALDI-TOF MS technique mainly relies on the comparison of mass spectra representing specific molecular fingerprints of microorganisms, mostly proteins. It is known that the composition of the growth medium may lead to changes in proteins expression reflected in the variation of molecular profiles what may affect the result of identification [43]. MSP spectra used for identification are a set of signals consisting mainly of conserved ribosomal proteins, which constitute about 50% of the sum of all signals [44]. The high proportion of ribosomal proteins implied high repeatability and reliability of bacterial species identification expressed by the Score values observed in our studies (Appendix A). This is clearly demonstrated by the examples of bacteria for which the highest Score values have been obtained, e.g., *M. morganii* cultured on VRE (Score value—2.48 ± 0.05), where signals 5382.4 *m/z*, 6353.4 *m/z*, 7275.3 *m/z*, that represented different ribosomal proteins (50S ribosomal protein L34, L32, and L29, respectively—according to the UniProt database), where detected in large amounts (Appendix A). Contrary to this, the composition of the medium significantly influences the expression of non-ribosomal bacterial proteins that are more or less metabolic status dependent, e.g., those responsible for drug resistance such as carbapenemases [45]. In our research, the greatest differences in identification levels were observed for species *C. striatum*, *M. morganii*, and *P. mirabilis* cultured on the BLA and VRE medium (Appendix A). For all mentioned species, the use of VRE medium gave significantly higher Scores values compared to the BLA—from 0.4 (*C. striatum*) to 0.7 (*M. morganii*) (Appendix A). The proteomic analysis of the mass spectra composition via UniProt database revealed many differences in the presence/absence of specific signals in mentioned bacterial species depending on the applied medium (Appendix A). Considering only characterized proteins, analysis disclosed from 11 to 59 signals that differentiate investigated bacteria regarding BLA and VRE media used. These changes included signals from both the lower mass range, e.g., 3094.3 *m/z* (Potassium-transporting ATPase subunit F), 5333.4 *m/z* (Heat shock protein J) or 6232.5 *m/z* (Lysophospholipase)—*M. morganii*, as well as those of higher *m/z*, such as 11684.1 *m/z* (Protein translocase subunit SecE) and 11707.4 *m/z* (DNA topoisomerase (ATP-hydrolyzing)—*C. striatum*. The noted phenomenon may result from the considerably different media compositions, such as the presence of defibrinated sheep blood and starch in the case of BLA as well as yeast extract, aesculin, ferric ammonium citrate, and sodium azide in VRE. A similar observation was noted in our previous work Złoch et al. 2020 [43], where the choice of the media types significantly influenced molecular profiles of different *S. aureus* strains leading to large variation in the efficiency of their strain-typing. It may result from the induction of various metabolic pathways in bacteria in response to specific media components. Additionally, some researchers pointed out that the use of blood-containing media poses a risk of contamination of the MS profiles with blood-related proteins [46]. Our findings suggest that although the use of all investigated culture media allowed to reliable bacteria identification at the species level, nevertheless, in the case of some bacterial species, the media selection may significantly improve the quality of the identification what was proved for *C. striatum*, *P. mirabilis*, and *M. morganii* cultured on the VRE medium. Therefore, during the evaluation of the usefulness of culture media for recovery microbial background of the DFI samples, besides selective properties of the culture media, also their influence on the molecular profiles of the bacteria should be concerned.

## 4. Materials and Methods

### 4.1. Clinical Samples

The superficial swabs were collected from infected diabetic foot wounds from 16 patients from Provincial Polyclinical Hospital in Toruń (Poland) using flocked swab (ESwab Collection System, Copan, Murrieta, CA, USA) according to the local guidelines by a specialist nurse applying the Levine technique. After wound debridement, samples for bacterial culture were obtained by swabbing the wound (the swab was rotated over a 1 cm^2^ area of the viable non-necrotic wound tissue). To avoid contaminating bacterial flora, wounds were debrided from necrotic or non-viable tissue or slough and rinsed with saline before swabbing. The samples were immediately placed into a liquid transport medium (Amies δswab, Deltalab, Nemours, France) and transported to Centre for Modern Interdisciplinary Technologies, where they were stored at −80 °C. The Ethical Committee approved the studies (Bioethical Commission’s permission no. 68/2019). Information about gender and age of the investigated DFI patients are provided in Appendix A.

### 4.2. Bacteria Isolation and Culturing Technique

After defrosting, samples were thoroughly vortexed and a series of 10-fold serial dilution (10^−1^ to 10^−3^) were prepared from them. For this purpose, 0.5 mL of the clinical sample was added to a test tube containing 4.5 mL of sterile peptone water (Sigma Aldrich, Steinhelm, Germany) and vortexed (first dilution—10^−1^). 100 μL of each dilution was plated onto 10 different types of culture media: Tryptic Soy Agar (TSA; Sigma Aldrich, Germany), Mueller Hinton Agar (MHA; Sigma Aldrich, Germany), Columbia Agar Base (COL; Oxoid, UK), Columbia Blood Agar (BLA; Oxoid, UK), Brain Heart Infusion Agar (BHI; Sigma Aldrich, Germany), CHROMagar Orientation (CHRA; GRASO Biotech, Poland), Azide Blood Agar (AZI; Oxoid, UK), Glucose Bromocresol Purple Agar (BCP; Sigma Aldrich, Germany), Mannitol Salt Agar (MAN; Oxoid, UK), Vancomycin-Resistant Enterococi Agar (VRE; Oxoid, UK). All media were in the form of ready-to-use powders except for BLA, which was prepared by adding defibrinated sheep blood (GRASO Biotech, Poland) to the sterilized and dissolved Colombia blood agar base to the final concentration 5% (*v*/*v*). After incubation at 37 °C for 18–24 h on the basis of morphological differences, single colonies of bacteria were selected, from which reduction cultures were prepared on the same media in order to obtain pure cultures (incubation at 37 °C for 18–24h).

### 4.3. Identification of Bacterial Isolates Using MALDI-TOF MS Technique

The common formic acid/acetonitrile method of protein extraction was used according to the protocol of the producer of the MALDI Biotyper system—the company Bruker Daltonik (Bremen, Germany). 1 inoculation loop (10 µL) of biomass was added to an Eppendorf tube containing 300 µL of sterile deionized water and mixed. Then, in order to inactivate viable bacterial cells, 900 μL of 96% ethyl alcohol was added to the suspension and mixed again followed common procedure recommended by manufacturer. The suspension containing rendered non-viable bacterial cells was centrifuged (1300 rev/min, 5 min), the supernatant was discarded and the remaining cell pellet was dried using a vacuum centrifuge at room temperature. Then 10 µL formic acid (FA) and 10 µL acetonitrile (ACN) was added to dried cell pellet. After mixing, the sample extract was centrifuged (13,000 rev/min, 5 min) and 1 µL of supernatant was transferred onto a MALDI MTP 384 ground steel target sample spot (Bruker Daltonik GmbH, Germany). After air-drying, the sample spot was overlaid with 1 µL of respective matrix solution α-cyano-4-hydroxycinnamic acid (HCCA; Sigma Aldrich, Switzerland): 10 mg/mL in standard solvent solution (50% ACN, 47.5% water and 2.5% trifluoroacetic acid).

Samples were analyzed using a ultrafleXtreme MALDI-TOF mass spectrometer (Bruker Daltonik GmbH, Bremen, Germany) equipped with the smartbeam-II laser–positive mode. Spectra were collected manually using manufacturer software, flexControl version 3.4 build 135 (parameters: 500 shots in-one-single spectra to frequency 2500, *m/z* range = 2000–20,000, acceleration voltage = 25 kV, global attenuator offset = 20% and attenuator offset = 34% and its range = 34%, laser power = 40%), and subjected to smoothing using the Savistsky–Golay method (width 2 *m/z*, cycles 10) and baseline corrections using the TopHat algorithm (signal to noise threshold 2; peak detection algorithm–centroid) followed by calibration using the Bruker’s Bacterial Test Standard (BTS; Bruker Daltonik, Bremen, Germany) in quadratic mode via manufacturer software, flexAnalysis version 3.4 build 76. Each sample was measured in quadruplicate (two spots per samples measured twice). Validated mass spectra were used for bacterial identification via MALDI Biotyper 4.1 Platform (Bruker Daltonik GmbH, Bremen, Germany) based on both raw spectra (RAW) and main spectra (MSP).

## 5. Conclusions

The introduction of the MALDI-TOF MS technique into the clinical laboratories caused the application of the culturomics approach for the fast and reliable analysis of infections to become meaningful. Still, the selection of the most suitable growth conditions is challenging since it requires extending the research to various geographic regions and a wider range of media. Performed studies proved that the choice of the suitable culture media for culturomics approach demonstrates considerable influence on both efficiency of reflecting the microbial composition of the DFI samples (selective properties of the media) as well as on the quality of the identification in the cases of some bacterial species (impact on the composition of bacterial molecular profiles). Our studies indicated that especially two media—chromagar orientation and vancomycinresistant enterococi agar, which have not yet been widely used for revealing the DFI microbial composition, demonstrated high utility in studying the microbial composition of the DFI in patients in Poland by means of culturomics approach. Further studies involving samples derived from other parts of Poland, Europe, and the rest of the world may help to justify whether its application for routine analysis will be beneficial.

## Figures and Tables

**Figure 1 ijms-22-09574-f001:**
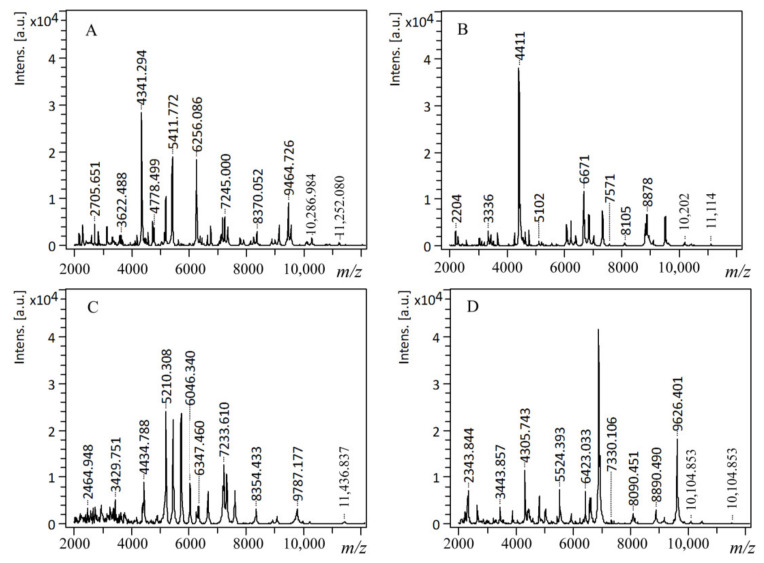
Exemplary mass spectra of bacterial protein extracts (**A**)-*Enterococcus faecalis*, (**B**)-*Klebsiella oxytoca*, (**C**)-*Pseudomonas aeruginosa*, (**D**)-*Staphylococcus aureus*) obtained during MALDI TOF/MS analysis using positive ion mode. Intens. [a.u.]: signals (peaks) intensities given in arbitrary units; *m/z*: mass-to-charge ratio, the ratio of an ion’s mass (m) in atomic mass units (amu) to its formal charge (z)-in the case of our study z= +1 due to the use of positive ionization mode.

**Figure 2 ijms-22-09574-f002:**
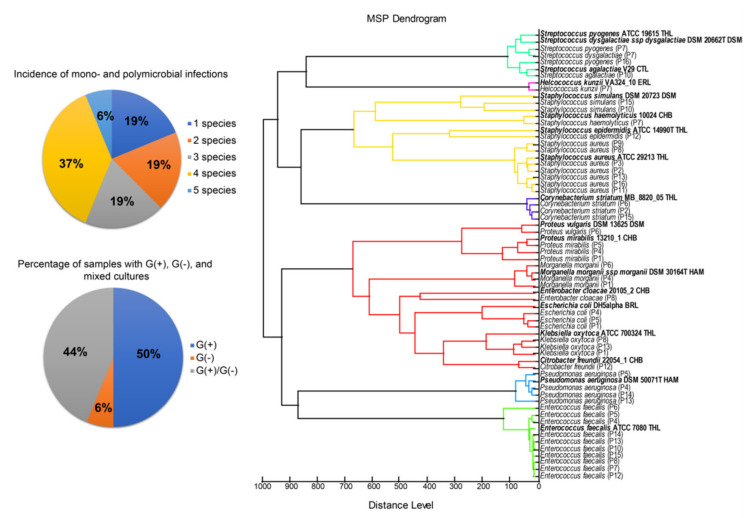
Phyloproteomic relationship between identified DFI bacterial species (MSP dendrogram) with diagrams showing percentage of mono- and polymicrobial infections as well as incidence of Gram-positive, Gram-negative, and mixed cultures among investigated samples.

**Figure 3 ijms-22-09574-f003:**
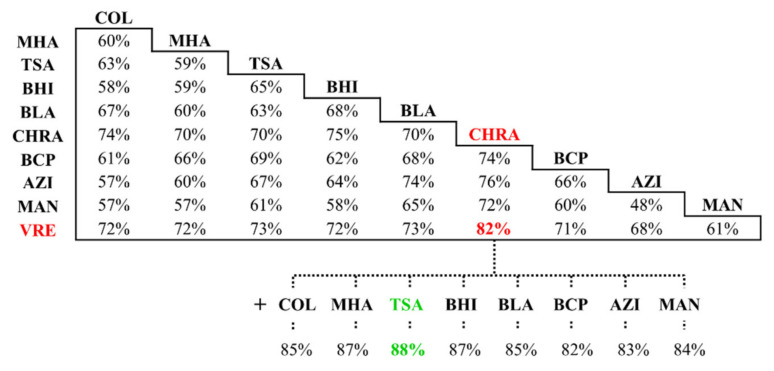
Selection of the most effective two-component and three-component sets of culture media based on the effectiveness of reflecting the microbial composition of the samples (expressed as a percentage of all identified bacterial species for the certain patient) using individual combinations of media. The most effective two-component set of media—CHRA + VRE (highlighted with red color), was selected for searching the most efficient three-component set (marked with the green color).

**Table 1 ijms-22-09574-t001:** Species coverage depending on culture media type used shown as percentage of identified isolates obtained using individual culture media.

Bacteria Species		Percentage of Identified Isolates Obtained Using Individual Culture Media									
		COL	MHA	TSA	BHI	BLA	CHRA	AZI	BCP	MAN	VRE
*Gram-positive*	*Corynebacterium striatum*	6%	5%	9%	5%	13%	11%	-	9%	-	8%
	*Enterococcus faecalis*	16%	16%	19%	19%	9%	18%	50%	18%	10%	28%
	*Streptococcus agalactiae*	5%	5%	5%	4%	-	3%	6%	5%	-	4%
	*Streptococcus dysgalactiae*	5%	5%	5%	5%	4%	-	-	4%	-	4%
	*Streptococcus pyogenes*	5%	5%	5%	5%	4%	3%	6%	5%	-	-
	*Staphylococcus aureus*	16%	21%	19%	24%	13%	7%	32%	18%	50%	20%
	*Staphylococcus epidermidis*	-	-	-	-	-	-	-	-	10%	-
	*Staphylococcus haemolyticus*	5%	5%	5%	5%	4%	3%	6%	4%	10%	4%
	*Staphylococcus simulans*	5%	-	-	-	4%	3%	-	-	-	4%
	*Helcococcus kunzii*	-	-	-	-	4%	-	-	-	-	4%
*Gram-negative*	*Escherichia coli*	-	-	-	-	-	11%	-	4%	-	-
	*Klebsiella oxytoca*	11%	11%	-	9%	8%	11%	-	9%	-	-
	*Citrobacter freundii*	-	5%	5%	5%	4%	4%	-	4%	-	4%
	*Enterobacter cloacae*	5%	-	5%	-	-	4%	-	-	-	-
	*Morganella morganii*	-	-	-	5%	4%	7%	-	4%	-	-
	*Proteus mirabilis*	11%	11%	9%	14%	13%	4%	-	9%	20%	8%
	*Proteus vulgaris*	-	-	-	-	4%	4%	-	4%	-	-
	*Pseudomonas aeruginosa*	5%	11%	14%	-	8%	7%	-	4%	-	8%

**Table 2 ijms-22-09574-t002:** Number of Gram-positive and -negative species and isolates obtained depending on the culture medium used.

Medium	Species			Isolates		
	G(+)	G(−)	Σ	G(+)	G(−)	Σ
COL	8	5	13	12	7	19
MHA	7	4	11	12	7	19
TSA	7	4	11	14	7	21
BHI	7	4	11	14	7	21
BLA	9	6	15	14	10	24
CHRA	7	8	15	14	14	28
AZI	5	0	5	16	0	16
BCP	7	7	14	14	9	23
MAN	4	1	5	8	2	10
VRE	8	4	12	19	6	25

**Table 3 ijms-22-09574-t003:** The usefulness of the culture media to reflect the microbial compositions of the tested DFI samples using the MALDI technique expressed as the percentage of all identified isolates in the individual patients. Total–percentage of identified isolates for all investigated patients; In 2 media set–average percentage of reflected microbial composition when the individual medium is used in combination with another one and calculated from values obtained for all 9 possible combinations (mean ± SD); In 3 media set–average percentage of reflected microbial composition when the individual medium is used in combination with two other media and calculated from values obtained for all possible combinations—36 (mean ± SD).

Patient	Reflected Microbiota Pattern [%]									
	COL	MHA	TSA	BHI	BLA	CHRA	BCP	AZI	MAN	VRE
P1	25	25	25	25	25	75	0	0	0	0
P2	50	50	50	100	50	50	50	0	0	50
P3	100	0	0	100	0	0	100	100	100	100
P4	20	20	0	20	20	60	40	20	20	60
P5	0	50	50	25	50	75	50	25	25	50
P6	0	25	25	25	75	75	50	25	0	0
P7	75	75	75	75	100	50	75	50	25	100
P8	75	50	75	50	75	75	50	25	25	50
P9	0	100	100	100	100	0	100	100	100	100
P10	100	67	67	67	33	67	67	67	33	67
P11	100	100	100	100	100	100	100	100	100	100
P12	33	33	33	33	33	33	33	33	33	67
P13	50	25	50	50	25	50	50	50	25	50
P14	50	50	50	50	50	50	50	50	0	50
P15	0	0	0	0	67	67	33	0	0	100
P16	50	50	100	50	50	50	50	50	0	0
Total	46	45	50	54	53	55	56	43	30	59
In 2 media set	63 ± 6	62 ± 5	66 ± 5	65 ± 6	67 ± 5	73 ± 4	66 ± 5	65 ± 8	60 ± 7	71 ± 5
In 3 media set	73 ± 4	73 ± 4	75 ± 4	74 ± 4	75 ± 3	81 ± 2	74 ± 3	76 ± 3	72 ± 4	78 ± 3

## Data Availability

Not applicable.

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
