# Peer review of "Culturomics Approach to Identify Diabetic Foot Infection Bacteria"

_ijms, 2021, doi:10.3390/ijms22179574_

Round 1
Reviewer 1 Report
In this manuscript, ZÅ‚och et al. applied the culturomics approach and combined it with assisted laser desorption ionization-time of flight mass spectrometry to identify the bacterial foot infection from 16 diabetic patients in Poland. This study utilizes a conventional approach to address a classic topic with broad relevance, and the data presented are logical. The study is helpful to promote the improvement of the culturomic approach. However, there are several limitations in this study that the author should be considering and the conclusions were drawn:
- TOF MS is a well-accepted tool for bacterial identification in the clinical microbiology study. The results in Figure 1 which are showing the standard mass spectra of total protein from different bacteria strain, it has been published (Carbonnella et al. 2011). It is inappropriate to use a published data as the main data of the manuscript. And it is not clear if the method in this manuscript is upgraded or have more advantage than the others. However, to assert this, the authors should describe the innovation of the strategy.
- The samples number (only 16 patients) is very limited in this study. The data of gender distribution is missed as well(Figure 2 and Table 3) . Generally, for a research object and population, at least 40 samples should be compared. If you consider the statistical difference of gender, you should have more than 80 people in total.
- Overall, this work is methodological research that focuses on the usefulness of the culturomics approach on the bacteriology of diabetic foot infections. The author concluded that two rarely used media, chromagar orientation, and vancomycin-resistant enterococci agar can be used in the DFI microbial composition study. However, These two media are commercially developed products and have been widely used in microbiology research. The conclusion drawn in this manuscript is lacking sufficient data to prove and innovative points.
- Looking at the data as a whole, the author mostly confirms or compare the result from the previous study. There is no in-depth discussion of the molecular mechanism of the research results.
Author Response
Response to Reviewers’ Comments
Ref. No.: ijms-1304799
Culturomics approach to identify diabetic foot infection bacteria
Michał Złoch, Ewelina Maślak, Wojciech Kupczyk, Marek Jackowski, Paweł Pomastowski and Bogusław Buszewski
Firstly, the authors would like to thank the Editor and Reviewers for appreciation of our effort and secondly, for useful comments, remarks and valuable suggestions that led to the increasing of quality of the present work. Therefore, to better improve the article the authors have addressed all the comments as explained below.
In manuscript file all of the changes have been done using the "Track Changes" function in Microsoft Word.
Reviewers' comments:
Reviewer #1
Comments and Suggestions for Authors
In this manuscript, ZÅ‚och et al. applied the culturomics approach and combined it with assisted laser desorption ionization-time of flight mass spectrometry to identify the bacterial foot infection from 16 diabetic patients in Poland. This study utilizes a conventional approach to address a classic topic with broad relevance, and the data presented are logical. The study is helpful to promote the improvement of the culturomic approach. However, there are several limitations in this study that the author should be considering and the conclusions were drawn:
- TOF MS is a well-accepted tool for bacterial identification in the clinical microbiology study. The results in Figure 1 which are showing the standard mass spectra of total protein from different bacteria strain, it has been published (Carbonnella et al. 2011). It is inappropriate to use a published data as the main data of the manuscript. And it is not clear if the method in this manuscript is upgraded or have more advantage than the others. However, to assert this, the authors should describe the innovation of the strategy.
Re: Thanks to the Reviewer for the remark. We completely agree with the Reviewer’s suggestion that the use of a published data as the main data is inappropriate, however, Figure 1 presented our original results which are mass spectra of 4 different bacterial species obtained during performed analysis. As proof of this, we have attached screenshots showing the raw spectra obtained during the analysis using FlexAnalysis software and were used for Figure 1 preparation. Although TOF MS is a well-accepted tool for bacterial identification, still the application of the MALDI approach for routine analysis in clinical microbiology is considered a relatively new method that revolutionized microbial identification. Therefore, in our opinion, it is important to attach a figure with the example of MS spectra obtained for investigated bacteria that will support understanding of presented findings, especially for a non-specialist. As for the innovation of the strategy presented in this manuscript, it is a use of fast and reliable MALDI TOF MS identification along with the multiplication of culture conditions – so called modern culturomics approach. Although in the current literature we can find examples of culturomics approach application for investigation of different clinical specimens, also DFI samples (e.g. Jneid et al. 2018, doi: 10.3389/fcimb.2018.00282), however, none of them, to our knowledge, provide detailed evaluation of the usefulness of specific media and their sets for the diagnosis of diabetic foot infection including impact on the molecular profiles of the identified bacteria. Moreover, our studies were focused on the culture media that are more or less commonly used in present clinical laboratories, therefore, our findings can be potentially easily incorporated into the routine laboratories workflow in the close future.
- The samples number (only 16 patients) is very limited in this study. The data of gender distribution is missed as well(Figure 2 and Table 3) . Generally, for a research object and population, at least 40 samples should be compared. If you consider the statistical difference of gender, you should have more than 80 people in total.
Re: Thanks to the Reviewer for the valuable suggestion. The collection of samples for this study was limited due to the SARS-CoV-2 virus pandemic situation, which implied the limited availability of patients suffering from diabetic foot infection. Nevertheless, the presented work is a kind of methodological research in which we focused on the investigation of the same samples in multiple conditions. All 16 DFI samples were cultured on 10 investigated media in triplicate. For each medium ~5 colonies that differed morphologically were chosen for further MALDI analysis, giving ca 1500 analyzed MS spectra in total. In our opinion, such big data set was enough to compare studied culture conditions and drawn reliable conclusions. Gender distribution was missed due to the limited number of patients, as Reviewer also pointed, and because it was out of main goal of this studies. Nonetheless, with revised version of the manuscript we provide Table S2 with information about gender and age of investigated patients.
- Overall, this work is methodological research that focuses on the usefulness of the culturomics approach on the bacteriology of diabetic foot infections. The author concluded that two rarely used media, chromagar orientation, and vancomycin-resistant enterococci agar can be used in the DFI microbial composition study. However, These two media are commercially developed products and have been widely used in microbiology research. The conclusion drawn in this manuscript is lacking sufficient data to prove and innovative points.
Re: Thanks the Reviewer for pointed out this important issue. In our studies we used intentionally 10 different commercially available culture media bearing in mind the possibility of easy incorporation of the obtained results for future applications in routine microbiological analyzes. Investigated culture media are more or less commonly used in the clinical laboratories around the world, however, mostly in the conventional way, that is, phenotypic identification based on the isolating and growing microbial colonies on selective or/and differential media. Such approach is very time-consuming and quality of the identification results (level of identification) is limited. These limitations can be overcome using MALDI TOF MS technique which enables identification many microbial colonies in very short time, therefore, culturing clinical specimens on the many different culture media at the same time to provide better recovery of vary bacterial species present in the samples is reasonable. Moreover, reliability of the identification is comparable with that of 16S rDNA sequencing. Other important aspect raised in our studies is that many different type of culture media for recovery bacteria from DFI samples are used which is given as one of the possible factor responsible for inconsistent results between different studies. Results of our studies give new light on the importance of the culture media selection for efficient recovery of DFI microbial background as well as on the role of the culture media composition on the changes in the molecular profiles of investigated bacteria what can impact on the identification results. To emphasize this, goals and conclusions of our studies were revised.
- Looking at the data as a whole, the author mostly confirms or compare the result from the previous study. There is no in-depth discussion of the molecular mechanism of the research results.
Re: Thanks to the Reviewer for the valuable suggestion. Having known that microbial identification via the MALDI-TOF MS technique mainly relies on the comparison of mass spectra representing specific molecular fingerprints of microorganisms and that the composition of the growth medium may lead to changes in proteins expression reflected in the variation of molecular profiles and further affect the result of microorganisms identification, we agree with the Reviewer’s opinion, that molecular aspect of this studies should be deepened. Therefore, in the revised version of the manuscript, we add an in-depth discussion of the molecular mechanisms coming from the research results, that is, the impact of the culture media composition on the molecular profiles of the investigated bacteria in relation to the obtained identification levels. Discussion of the mentioned mechanisms was performed based on the prepared MS profiles presented in the Figure S1 as well as Tables S1 and S3 containing information about Score values of identifications depending on the culture media used as well as about proteins that differentiate some bacterial strains depending on the culture media (characterized using the UniProt database). Mentioned Tables and Figure are provided in Supplementary Materials.

Reviewer 2 Report
Culturomics is a high throughput method that has enabled the isolation of large numbers of species from individual samples or specimens. Here, swab samples from the foot infections of 16 diabetic patients were cultured for the bacterial populations present. Great differences in the species composition were observed. The authors concluded that a culturomics approach improves the accuracy of microbial composition determinations for diabetic foot infections.
Microbiological composition analysis was performed on swab samples obtained from 16 diabetic foot infection patients using MALDI-TOF MS. 204 bacterial isolates were obtained representing 18 different species. The yield of Gram-positive and Gram-negative species using varied culture media was described. Big differences between the efficiency of the various culture media in faithfully reflecting the microbial composition were observed. Selection of the most suitable growth conditions remains a challenge.
This was an interesting study, and it was clearly described. It deserves to be published with moderate revisions for sentence structure and grammar.
Author Response
Response to Reviewers’ Comments
Ref. No.: ijms-1304799
Culturomics approach to identify diabetic foot infection bacteria
Michał Złoch, Ewelina Maślak, Wojciech Kupczyk, Marek Jackowski, Paweł Pomastowski and Bogusław Buszewski
Firstly, the authors would like to thank the Editor and Reviewers for appreciation of our effort and secondly, for useful comments, remarks and valuable suggestions that led to the increasing of quality of the present work. Therefore, to better improve the article the authors have addressed all the comments as explained below.
In manuscript file all of the changes have been done using the "Track Changes" function in Microsoft Word.
Reviewers' comments:
Reviewer #2
Comments and Suggestions for Authors
Culturomics is a high throughput method that has enabled the isolation of large numbers of species from individual samples or specimens. Here, swab samples from the foot infections of 16 diabetic patients were cultured for the bacterial populations present. Great differences in the species composition were observed. The authors concluded that a culturomics approach improves the accuracy of microbial composition determinations for diabetic foot infections.
Microbiological composition analysis was performed on swab samples obtained from 16 diabetic foot infection patients using MALDI-TOF MS. 204 bacterial isolates were obtained representing 18 different species. The yield of Gram-positive and Gram-negative species using varied culture media was described. Big differences between the efficiency of the various culture media in faithfully reflecting the microbial composition were observed. Selection of the most suitable growth conditions remains a challenge.
This was an interesting study, and it was clearly described. It deserves to be published with moderate revisions for sentence structure and grammar.
Re: Thanks the Reviewer for the revision and suggestions. The manuscript has undergone revision by an English native speaker to improve sentence grammar and structure.
Round 2
Reviewer 1 Report
Thanks for the corrections and clarification that the author has made, which improved the manuscript significantly. The discussion is much better especially. There are still a few minor edits:
1. Since the author wants to show figure 1 as the main result here, the figure legend should be improved to describe the detail of the parameters. It will be easier for the general reader to understand. For example, explain intens. and M/Z value showing on the y and axis.
2 .page 10 lines 9 and 10, C. striatum, P. mirabilis, and M. morganii should be Italic.
3. page 10, in 4.3 Identification of bacterial isolates using MALDI-TOF MS technique. The unit "ul" should be "uL". Please check all of it carefully.
Author Response
Response to Reviewers’ Comments
Ref. No.: ijms-1304799
Culturomics approach to identify diabetic foot infection bacteria
Michał Złoch, Ewelina Maślak, Wojciech Kupczyk, Marek Jackowski, Paweł Pomastowski and Bogusław Buszewski
The authors would like to thank the Editor and Reviewers for useful comments and valuable suggestions that led to the increasing quality of the present work. Therefore, to better improve the article the authors have addressed all the comments as explained below.
In manuscript file all of the changes have been done using the "Track Changes" function in Microsoft Word.
Reviewers' comments:
Reviewer #1
Comments and Suggestions for Authors
Thanks for the corrections and clarification that the author has made, which improved the manuscript significantly. The discussion is much better especially. There are still a few minor edits:
- Since the author wants to show figure 1 as the main result here, the figure legend should be improved to describe the detail of the parameters. It will be easier for the general reader to understand. For example, explain intens. and M/Z value showing on the y and axis.
Re: Thanks to the Reviewer for the valuable suggestion. In the revised version of the manuscript, we improve Figure 1 caption and legend according to the Reviewer's comment - "Exemplary mass spectra of bacterial protein extracts (A - Enterococcus faecalis, B - Klebsiella oxytoca, C - Pseudomonas aeruginosa, D - Staphylococcus aureus) obtained during MALDI TOF/MS analysis using positive ion mode. Intens. [a.u.]: signals (peaks) intensities given in arbitrary units; m/z: mass-to-charge ratio, the ratio of an ion's mass (m) in atomic mass units (amu) to its formal charge (z) - in the case of our study z= +1 due to the use of positive ionization mode". We believe that the revised legend of Figure 1 will help better understand the presented results.
2. page 10 lines 9 and 10, C. striatum, P. mirabilis, and M. morganii should be Italic.
Re: We would like to thank the Reviewer for such careful revision of the manuscript. Mentioned names of the bacteria have been corrected, that is, in the revised version of the manuscript they have been written in italic
- page 10, in 4.3 Identification of bacterial isolates using MALDI-TOF MS technique. The unit "ul" should be "uL". Please check all of it carefully.
Re: We would like to thank Reviewer for this remark. The unit "ul" has been corrected into "uL". Moreover, all the text has been carefully checked in terms of proper units names and all mistakes have been corrected.